# Structure of a type IV pilus machinery in the open and closed state

Vicki AM Gold[1]*, Ralf Salzer[2], Beate Averhoff[2], Werner Kühlbrandt[1]

[1]Department of Structural Biology, Max Planck Institute of Biophysics, Frankfurt am Main, Germany; [2]Molecular Microbiology and Bioenergetics, Institute of Molecular Biosciences, Goethe University Frankfurt, Frankfurt am Main, Germany

**Abstract** Proteins of the secretin family form large macromolecular complexes, which assemble in the outer membrane of Gram-negative bacteria. Secretins are major components of type II and III secretion systems and are linked to extrusion of type IV pili (T4P) and to DNA uptake. By electron cryo-tomography of whole *Thermus thermophilus* cells, we determined the *in situ* structure of a T4P molecular machine in the open and the closed state. Comparison reveals a major conformational change whereby the N-terminal domains of the central secretin PilQ shift by ∼30 Å, and two periplasmic gates open to make way for pilus extrusion. Furthermore, we determine the structure of the assembled pilus.

## Introduction

Secretins form multimeric pores through the outer membrane of Gram-negative bacteria (*Averhoff, 2009*; *Korotkov et al., 2011*; *Burkhardt et al., 2012*). They are the central secretion conduits for proteins and virulence factors in type II and III secretion systems (T2SS/T3SS) and are essential for extrusion of type IV pili (T4P) (*Martin et al., 1993*) and transport of some bacteriophages (*Korotkov et al., 2011*). In addition, secretins are key components of DNA transport systems, which mediate uptake of free DNA from the environment, referred to as natural transformation (*Schwarzenlander et al., 2009*). The ability to take up DNA is one of the major mechanisms of horizontal gene transfer (*Domingues et al., 2012*) and enables organisms to adapt rapidly to changing environments (*Averhoff, 2009*). This process is also fundamental for adaptation of pathogenic bacteria to human hosts and the acquisition of multi-drug resistance (*Domingues et al., 2012*).

In many Gram-negative bacteria, such as *Thermus*, and also in many major human pathogens such as *Neisseria*, *Pseudomonas*, and *Vibrio*, DNA uptake is linked to the T4P machinery (*Wolfgang et al., 1998*; *Graupner et al., 2000*; *Seitz and Blokesch, 2013*) (*Figure 1*). To investigate the structure and function of this system, we chose the thermophilic bacterium *Thermus thermophilus* HB27, which exhibits the highest transformation rates known to date (*Koyama et al., 1986*), and due to the thermostability of its proteins is a convenient model for structural studies.

Pili are several micron-long flexible filaments (*Craig and Li, 2008*) that can generate forces of over 100 pN (*Maier et al., 2004*). T4P are grouped together in a class based on the production and secretion of the major pilin protein PilA4 (*Thermus* nomenclature), thousands of copies of which form the helical pilus (*Craig et al., 2004*; *Schwarzenlander et al., 2009*). The T4P is the only known bacterial pilus that can be retracted rapidly (*Maier et al., 2004*) to enable motility and adherence (*Merz et al., 2000*), major contributors to bacterial virulence (*Hahn, 1997*). Assembly and disassembly of the pilus is driven by the AAA-ATPases (ATPases associated with diverse cellular activities) PilF (extension) and PilT1/PilT2 (retraction) (*Salzer et al., 2014b*). It has been suggested that mature PilA4 assembles into pili extending from the inner membrane by action of PilF (*Collins et al., 2013*; *Salzer et al., 2014b*). The outer membrane channel of the T4P machinery is formed by the

*For correspondence: vicki.gold@biophys.mpg.de

**eLife digest** Gram-negative bacteria can cause serious diseases in humans, such as cholera and bacterial meningitis. These bacteria are surrounded by two membranes: an inner membrane and an outer membrane. Proteins called secretins are components of several large molecular complexes that are embedded within the outer membrane. Some secretin-containing complexes form pores in the bacterial membranes and allow molecules to pass in or out of the cell.

Some secretins also form part of the machinery that allow Gram-negative bacteria to grow fibre-like structures called type IV pili. These pili help bacteria that cause infections to move and stick to host cells, where they can also trigger massive changes in the host cells' architecture. Multiple copies of a secretin protein called PilQ form a channel in the outer membrane of the bacteria that allows a type IV pilus to grow out of the surface of the cell. The pilus can then hook the bacteria onto surfaces and other cells. There is evidence to suggest the type IV pilus machinery is involved in the uptake of DNA from other bacteria, an important but poorly understood process that has contributed to the spread of multi-drug resistance.

Now, Gold et al. have used a cutting-edge technique called 'electron cryo-tomography' to analyse the three-dimensional structure of the machinery that builds the type IV pili in the membranes of a bacterium called *Thermus thermophilus*. This analysis revealed that, similar to many other channel complexes, the PilQ channel can be 'open' or 'closed'. When pili are absent, the channel is closed, but the channel opens when pili are present. Further analysis also revealed the structure of an assembled pilus.

Next, Gold et al. studied the open state of the type IV pilus in more detail and observed that a region of each of the PilQ proteins moves a considerable distance to make way for the pilus to enter the central pore. These results will pave the way for future studies of type IV pili and other secretin-containing complexes and underpin efforts to investigate new drug targets to combat bacterial infections.

dodacemeric ~1 MDa secretin complex PilQ (*Burkhardt et al., 2011*) (*Figure 1*). Other proteins, in particular PilM, PilN, and PilO, are hypothesized to be a central part of the pilus assembly platform and may couple the cytoplasmic and periplasmic sides of the T4P machinery (*Karuppiah et al., 2013*). Some proteins of the complex have been implicated to play a dual role in both pilus assembly and natural competence (*Friedrich et al., 2002*; *Averhoff and Friedrich, 2003*; *Friedrich et al., 2003*; *Rumszauer et al., 2006*). Recent results indicate that T4P themselves are not directly involved in DNA uptake (*Burkhardt et al., 2012*; *Salzer et al., 2014a*).

T4P are essential for pathogenesis by mediating adhesion, biofilm formation, and twitching motility (*Burrows, 2012*). Thus, both secretins and T4P play important roles in virulence of different pathogenic bacteria, which has fostered their use as new targets for drug development (*Baron, 2010*). To date, there is no information on the *in situ* structure of either the T4P machinery or DNA translocator. Determining structures of T4P complexes in whole bacterial cells is therefore of paramount importance and will enable further study of bacterial resistance and disease. Electron cryo-tomography (cryoET) has the unique ability to determine protein structures in cells at molecular resolution. We have applied cryoET to whole *T. thermophilus* HB27 cells, in order to visualize the T4P machinery *in situ*. We determine the helical structure of the pilus and find that the secretin complex PilQ is a central dynamic component of this system. CryoET and subtomogram averaging of the T4P machinery with and without pili reveal a ~30 Å conformational change as the gates in the complex open.

## Results and discussion

*T. thermophilus* has an unusual cell architecture with deep surface clefts, formed by invaginations of the outer membrane (*Figure 2A*). By cryoET, these clefts are seen to be constrictions that run around the cell body (*Figure 2D*). Distal to the most polar outer membrane ring, numerous fibrous and straight pili extend from the cell (*Figure 2A–D*). The pili are clearly associated with large protein complexes crossing the ~70 nm periplasm (*Figure 2B,C*). This distribution is in line with previous fluorescence and electron microscopy data, which demonstrate the polar localization of PilQ (*Seitz and Blokesch, 2013*) and pili (*Salzer et al., 2014c*).

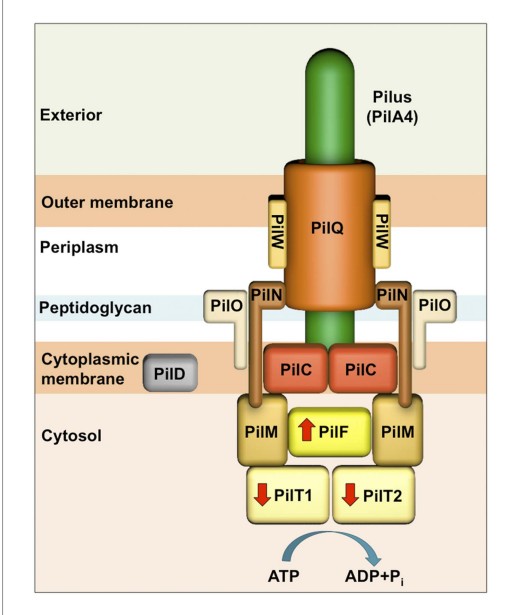

**Figure 1**. Schematic of the T4P machinery in *T. thermophilus*. The type IV pilus machinery is a heter-ooligomer, formed from at least 10 different pro-teins. The PilQ secretin (orange) forms a channel in the outer membrane for secretion of the pilus-forming protein PilA4 (green), which is processed by the prepilin peptidase PilD (grey) (*Friedrich et al., 2002*; *Schwarzenlander et al., 2009*). The mem-brane protein PilW (light orange) plays a role in DNA transport, PilQ assembly, and pilus extrusion (*Rumszauer et al., 2006*; *Schwarzenlander et al., 2009*). The dimeric complex PilC (red) is located in the inner membrane and is essential for pilus formation (*Friedrich et al., 2002*; *Karuppiah et al., 2010*). PilM (light brown), PilN (dark brown), and PilO (beige) are suggested to form the inner membrane assembly platform and connect the periplasmic and cytoplasmic sides of the complex (*Rumszauer et al., 2006*; *Schwarzenlander et al., 2009*; *Karuppiah and Derrick, 2011*; *Karuppiah et al., 2013*). The cytoplasmic ATPases PilF (bright yellow) and PilT1/PilT2 (pale yellow) drive pilus extension and retraction, respectively (indicated with red arrows) (*Rose et al., 2011*; *Salzer et al., 2014b*).

The periplasm of *T. thermophilus* cells is extraordinarily wide (*Quintela et al., 1995*; *Castan et al., 2002*) and too dense to select subvolumes for subtomogram averaging reliably. Therefore, cells were treated with 100 mM ethylenediaminetetraacetic acid (EDTA) and pipetted in order to render the outer membrane leaky (*Caston et al., 1988*). This had the desired effect of depleting the periplasm of most small proteins. Sample preparation by this method also removed the pilus from the complex, the empty T4P machinery was nonetheless still clearly visible (*Figure 3A,B*). We determined by subtomogram averaging the structure of the entire complex and found features distinct from those of the T2SS and T3SS secretins (*Marlovits et al., 2006*; *Hodgkinson et al., 2009*; *Reichow et al., 2010*). The resolution obtained by averaging ~4000 particle subvolumes was ~35 Å (*Figure 3—figure supplement 1*), most likely limited by the inherent flexibility of the complex (*Burkhardt et al., 2011*) and the difficulty of correcting precisely the contrast transfer function (CTF) for thick specimens.

Subtomogram average maps show the central protein channel (~35 nm long and ~15 nm wide) made up of several ring-shaped domains inserted into the outer membrane (*Figure 3C*, left panel). We compared a 2D projection of the channel part of the subtomogram average with projections of purified, negatively stained PilQ only (*Burkhardt et al., 2011*) (*Figure 3C*, central and right panels). PilQ was seen to consist of a C-terminal trapezoid 'cone structure' with staggered rings in the N-terminal domain (*Burkhardt et al., 2011*), in excellent agreement with our *in situ* structure. Moreover, the new cryoET structure shows additional protein densities extending from the putative N0 domain of PilQ through the pepti-doglycan layer (P1 and P2) to the cytoplasmic membrane (C1) (*Figure 3D*). Candidate proteins include PilW, which is associated with the inner and outer membranes and is essential for the outer membrane localization of PilQ (*Rumszauer et al., 2006*), and PilO/PilN heterodimers that could connect PilQ to the ATPases by PilM in the

cytoplasm (*Karuppiah and Derrick, 2011*; *Karuppiah et al., 2013*) (*Figure 1*). These proteins are most likely connected to one another by flexible domains that are not well contrasted in the subtomogram average. A longitudinal slice through the complex reveals that PilQ has two gates, which are closed in the absence of a pilus (*Figure 3C*, left panel and *Figure 3D*, right panel). Gate 1 is formed by the 'cone' in the outer membrane and gate 2 by the N1 domain at the base of PilQ, enclosing an empty periplasmic vestibule (*Figure 3D*, right panel). The C-terminal 'cone' has been shown to form a sodium dodecyl sulfate (SDS)-stable sub-domain (*Burkhardt et al., 2011*), thus, it is plausible that gate 1 is responsible for maintaining the integrity of the cell membrane in the closed state. A second gate formed by the N-terminal domains has not been observed in other secretins (*Korotkov et al., 2011*). The N-terminus of PilQ forms an unusual βββαββ fold, different from the

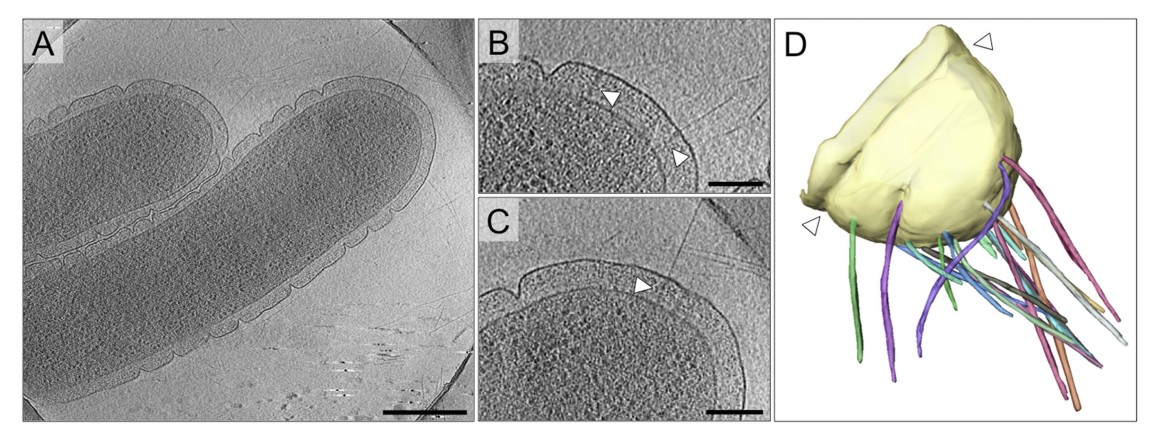

**Figure 2**. Cell morphology and pili of *T. thermophilus*. (**A–C**) Tomographic slices through *T. thermophilus* cells show invaginations in the outer membrane and large protein complexes crossing the periplasm (white arrowheads), which are associated with pili. Scale bar = 500 nm in **A**, 100 nm in **B** and **C**. (**D**) Volume rendering shows the distribution of pili (multi-coloured), protruding from the outer membrane (pale yellow). A concentric invagination of the outer membrane is indicated (white arrowheads).

conserved ring-building βαββα folds (*Burkhardt et al., 2012*). Thus, we hypothesize that this motif may form part of gate 2.

To determine the structure of the T4P machinery in the open state, the pipetting step during sample preparation was omitted ('Materials and methods'), which reduced the shearing forces and kept the assembled pilus intact (*Figure 4A,B*). Close inspection of the pilus shows a periodic structure (*Figure 4C*), which was suitable for subtomogram averaging. Since the ice surrounding the pili was thin (~100 nm), it was possible to apply CTF-correction. The structure of the pilus was determined at ~32 Å resolution by averaging 740 subvolumes (*Figure 3—figure supplement 1*). Power spectra calculated from a single tomographic slice or from the subtomogram average revealed a repeat distance of ~4.9 nm (*Figure 4C*, lower panels). The *T. thermophilus* pilus forms a right-handed helix of ~3 nm diameter (*Figure 4D*), which is different from the previously determined electron cryo-microscopy structure of the isolated *Neisseria gonorrhoeae* T4P (*Craig et al., 2006*). Diameters of pili can vary considerably and pilin proteins have limited sequence similarity (*Craig and Li, 2008*), which likely accounts for this difference. The result may prompt a reassessment of the functional roles of different T4P in cells. Due to the more challenging sample preparation method, the structure of the open T4P machinery with the pilus extended was determined from only ~300 particles at ~45 Å resolution (*Figure 3—figure supplement 1*). This resolution is sufficient to reveal the central protein channel complex with the ~3 nm pilus protein density in the centre (*Figure 4E*). An extensive conformational change is evident that shifts PilQ domains N0–N3 away from N4/N5 by ~30 Å towards the cytoplasmic membrane. This change opens the periplasmic vestibule to make way for the pilus (*Figures 4E, 5A,B* and *Video 1*). The shape and dimensions of the pore change from a tapered form ranging in width from ~4 nm (at N5) to ~8 nm (at N2/N3) in the closed state (*Figure 3C*), to a roughly constant 7 nm-wide channel in the open state (*Figure 4E*). In comparison, the closed state of *Neisseria meningitidis* PilQ is ~9 nm wide tapering to a point (*Collins et al., 2004*), which would be sufficient to accommodate the wider *Neisseria* pilus (*Collins et al., 2003*; *Craig et al., 2006*) after a comparably large conformational change. Additional conformational changes and shifts also occur in protein densities P1, P2, and C1 (*Figure 5A*). In the open state, we observe an extra protein density in the cytoplasm (*Figure 4E*, yellow arrowheads and *Video 1*), which we speculate could be PilF, linked to the inner membrane platform via PilM (*Karuppiah and Derrick, 2011*) (*Figure 1*). Homologous proteins have been shown to interact with the cytoplasmic ATPases, stimulating their activity (*Lu et al., 2013*, *2014*). Because EDTA treatment caused a depletion of periplasmic protein, we cannot exclude the possibility that some proteins may have been removed from the T4P machinery. However, we also averaged complexes from a tomogram of the two cells shown in *Figure 2*, which contained both open (with pili) and closed (without pili)

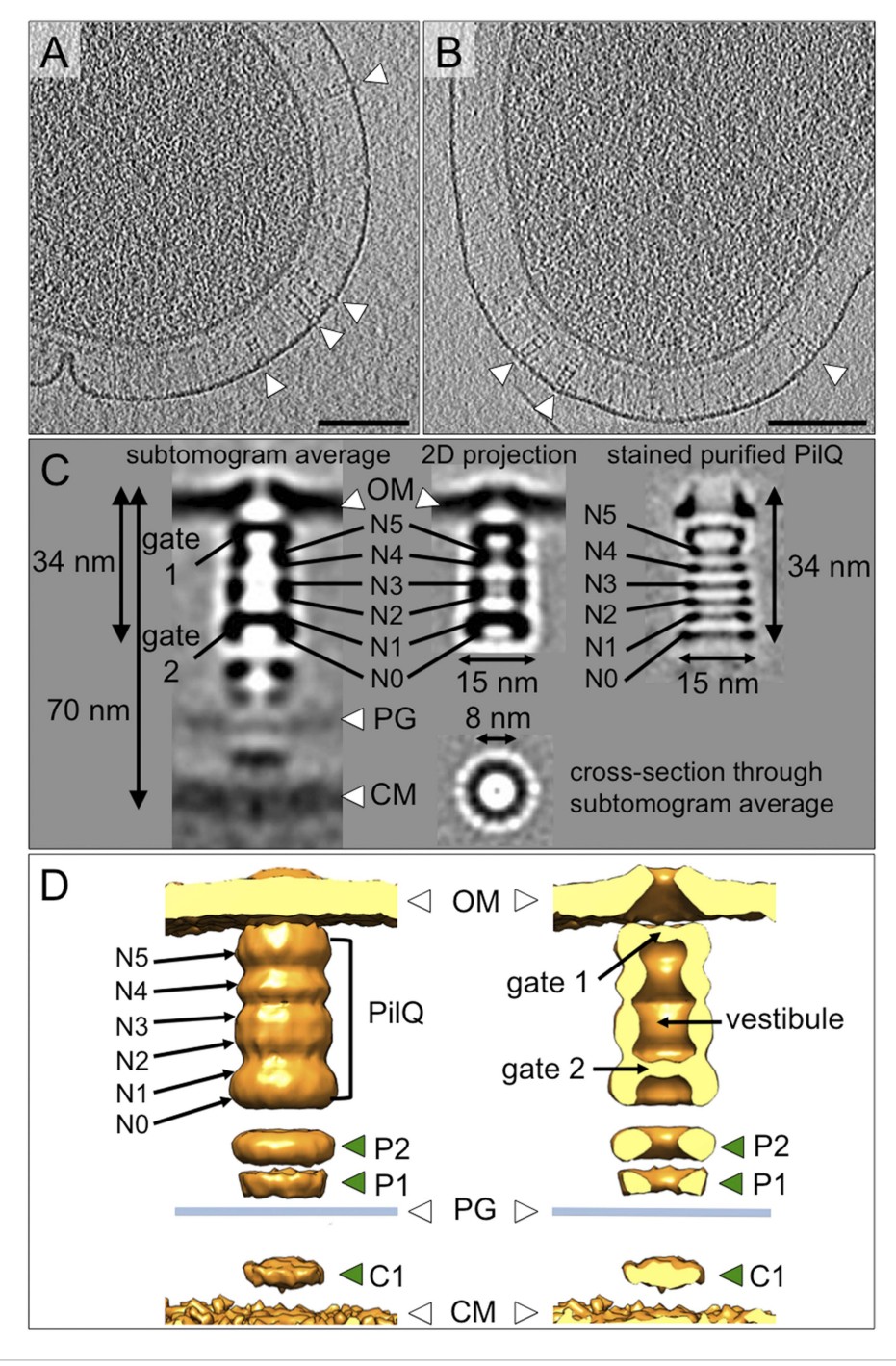

**Figure 3**. Structure of the T4P machinery in the closed state. (**A** and **B**) Tomographic slices of *T. thermophilus* cells show large protein complexes crossing the periplasm in the absence of pili (white arrowheads). Scale bars = 100 nm. (**C**) Resulting subtomogram average (left panel) and its 2D projection (centre) are compared to the previously determined projection map of isolated and stained PilQ (right panel) (**Burkhardt et al., 2011**). The contrast of the stained PilQ has been inverted. This image was originally published in The Journal of Biological Chemistry. Janin Burkhardt, Janet Vonck, and Beate Averhoff. Structure and Function of PilQ, a Secretin of the DNA Transporter from the Thermophilic Bacterium *T. thermophilus* HB27. *JBC*. 2011; 286:9977–9984, the American Society for Biochemistry and Molecular Biology. The putative N0–N5 domains of PilQ (**Burkhardt et al., 2012**) are marked. (**D**) 3D surface rendering of the average reveals

*Figure 3. continued on next page*

*Figure 3. Continued*

that PilQ has a periplasmic vestibule closed at both ends by two gates. Additional protein densities distinct from PilQ (green arrowheads; C1 = proximal to the cytoplasmic membrane, P1 = central periplasmic ring 1, P2 = central periplasmic ring 2) are also shown. OM, outer membrane; PG, peptidoglycan; CM, cytoplasmic membrane.

The following figure supplement is available for figure 3:

**Figure supplement 1**. Fourier shell correlation curves for subtomogram averages.

complexes and where the periplasmic protein density was considerably higher. These subtomogram averages (*Figure 5—figure supplement 1*) show clearly that the same large conformational changes occur in the T4P machinery, irrespective of the degree of periplasmic protein depletion, and hence of the effect of EDTA.

Docking the structures of the closed complex back into the tomographic volume reveals their three-dimensional distribution in the cell (*Figure 5C,D* and *Video 2*). We find that the T4P machinery tends to be tightly packed, with an inter-particle distance of 30–40 nm (*Figure 5E*). When the pili were depleted by pipetting, each cell contained on average 33 ± 19 closed complexes. However, *T. thermophilus* assembles ~6 pili per cell (*Salzer et al., 2014a*, *2015*), suggesting that ~80% of the complexes are not involved in pilus formation under standard growth conditions (*Salzer et al., 2014c*). We speculate that these idle complexes may form a second class of transporter, which may be active in DNA uptake or protein secretion.

Taken together, our results demonstrate that the DNA translocator protein PilQ forms a dynamic central component of the T4P machinery in *T. thermophilus*. The length of the complex across the wide periplasm may be an adaptation to the thermophilic environment that *Thermus* thrives in. However, core components of the T4P machinery are conserved in bacteria (*Nudleman and Kaiser, 2004*), and thus, we speculate that the overall architecture may be similar. Our findings will enable further structure-function studies of the proteins that comprise this elaborate and important macromolecular machine.

## Materials and methods

### Strains and culture conditions

*T. thermophilus* HB27 was grown in TM$^+$ medium (8 g/l tryptone, 4 g/l yeast extract, 3 g/l NaCl, 0.6 mM MgCl$_2$ 0.17 CaCl$_2$) (*Oshima and Imahori, 1971*). Cells from a 24-hr pre-culture were transferred onto TM$^+$ plates (containing 2% [wt/vol] agar) and incubated under humid conditions for 48 hr at 68°C.

### Sample preparation

To determine the structure of the closed complex, cubes of agar with growing *T. thermophilus* cells were cut out and placed into buffer containing 20 mM Tris pH 7.4, 100 mM EDTA and gently agitated for 1 hr at room temperature. Samples were mixed 1:1 with 10 nm protein A-gold (Aurion, Wageningen, The Netherlands) as fiducial markers and applied to glow-discharged R2/2 Cu 300 mesh holey carbon-coated support grids (Quantifoil, Jena, Germany) by gentle pipetting. For the structure of the open complex, cells were treated with EDTA as above, then protein A-gold was added and grids dipped into the solution without the pipetting step. Grids were blotted for ~4 s in a humidified atmosphere and plunge-frozen in liquid ethane in a home-made device. Grids were maintained under liquid nitrogen and transferred into the electron microscope at liquid nitrogen temperature.

### CryoET

Tomograms were typically collected from +60° to −60° at tilt steps of 2° and 5–7 µm underfocus, using either a Tecnai Polara or Titan Krios microscope (FEI, Hillsboro, USA), both equipped with field-emission guns operating at 300 kV and Quantum energy filters (Gatan, Pleasanton, USA) operated at a slit width of 20 eV. Both instruments were fitted with K2 Summit direct electron detector cameras (Gatan, Pleasanton, USA). Dose fractionated data (3–5 frames per projection image) were collected using Digital Micrograph (Gatan, Pleasanton, USA) at a nominal magnification of 34,000× (corresponding to a pixel

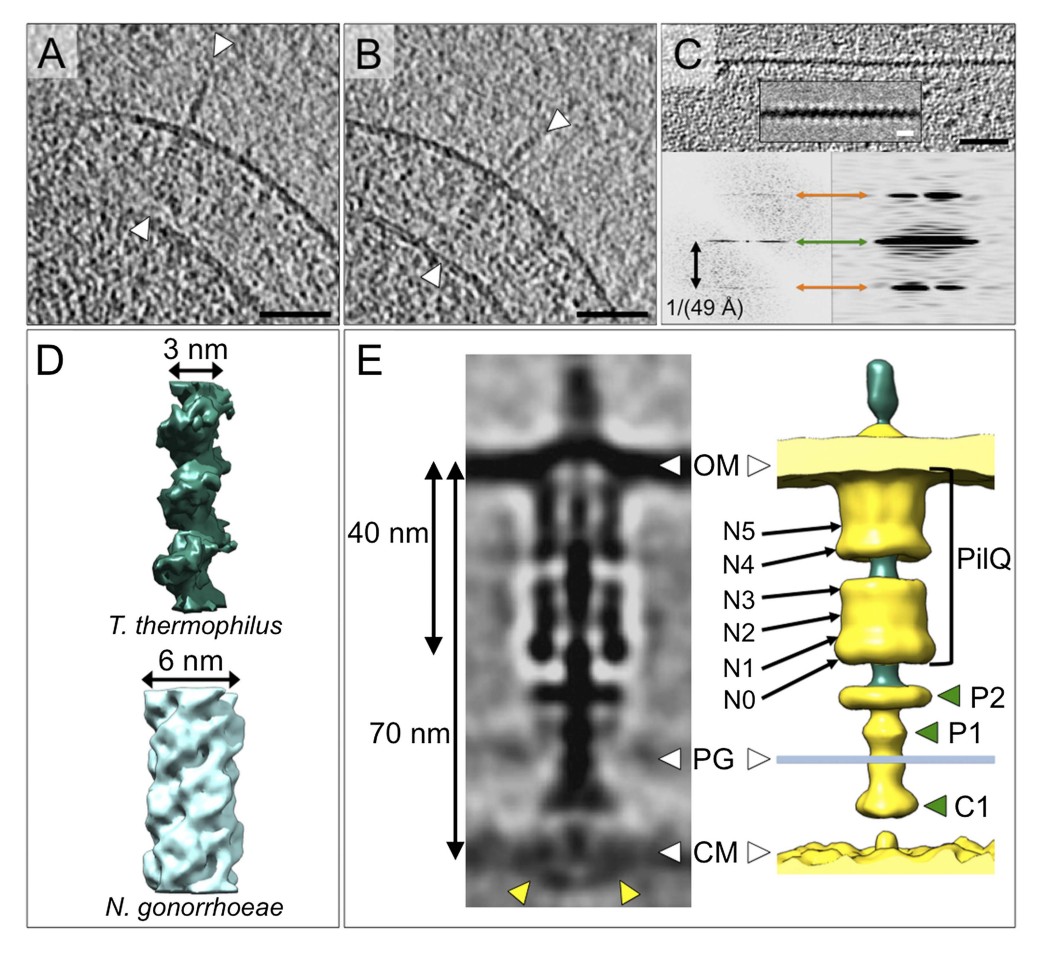

**Figure 4**. Structure of the T4P and the assembled machinery in the open state. (**A** and **B**) Tomographic slices show close-up views of the T4P machinery with assembled pili (white arrowheads). Scale bar = 50 nm. (**C**) Upper panel, a tomographic slice of the pilus shows that the structure is periodic. Scale bar = 20 nm. A slice through the subtomogram average (inset) shows the repeat more clearly. Scale bar = 5 nm. Lower panels, power spectra of the tomographic slice (left) and the average (right) depict layer lines (orange arrows) at a distance of 1/(49 Å) from the equator (green arrow), corresponding to a helical pitch of 4.9 nm. The contrast has been inverted. (**D**) Subtomogram average of the ~3 nm wide *T. thermophilus* pilus (green, top panel), compared to the previously determined ~6 nm wide structure from *N. gonorrhoeae* (blue, bottom panel, EMDB 1236) (*Craig et al., 2006*). (**E**) Subtomogram average (left panel) and 3D surface rendering (right panel) of the T4P machinery in the open state with the central pilus (green). The putative N0–N5 (*Burkhardt et al., 2012*) domains of PilQ are marked. Additional protein densities distinct from PilQ (green arrowheads; C1 = proximal to the cytoplasmic membrane, P1 = central periplasmic ring 1, P2 = central periplasmic ring 2) are also shown. Compared to the closed state of the complex, additional protein densities (left panel, yellow arrowheads) are visible in the cytoplasm. See *Video 1* for details. OM, outer membrane; PG, peptidoglycan; CM, cytoplasmic membrane.

size of 0.66 nm) in the Polara or at 33,000× (corresponding to a pixel size of 0.42 nm) in the Krios. The total dose per tomogram was ~140e⁻/Å². Tomograms were aligned using gold fiducial markers and volumes reconstructed by weighted back-projection using the IMOD software (Boulder Laboratory, Boulder, USA) (*Kremer et al., 1996*). Contrast was enhanced by non-linear anisotropic diffusion (NAD) filtering in IMOD (*Frangakis and Hegerl, 2001*). Segmentation was performed using AMIRA (FEI, Hillsboro, USA).

## Subtomogram averaging

Data collected at 34,000× and 8 μm underfocus on the Tecnai Polara were used to calculate the subtomogram averages shown in *Figure 5—figure supplement 1*. Subtomogram averages of the T4P

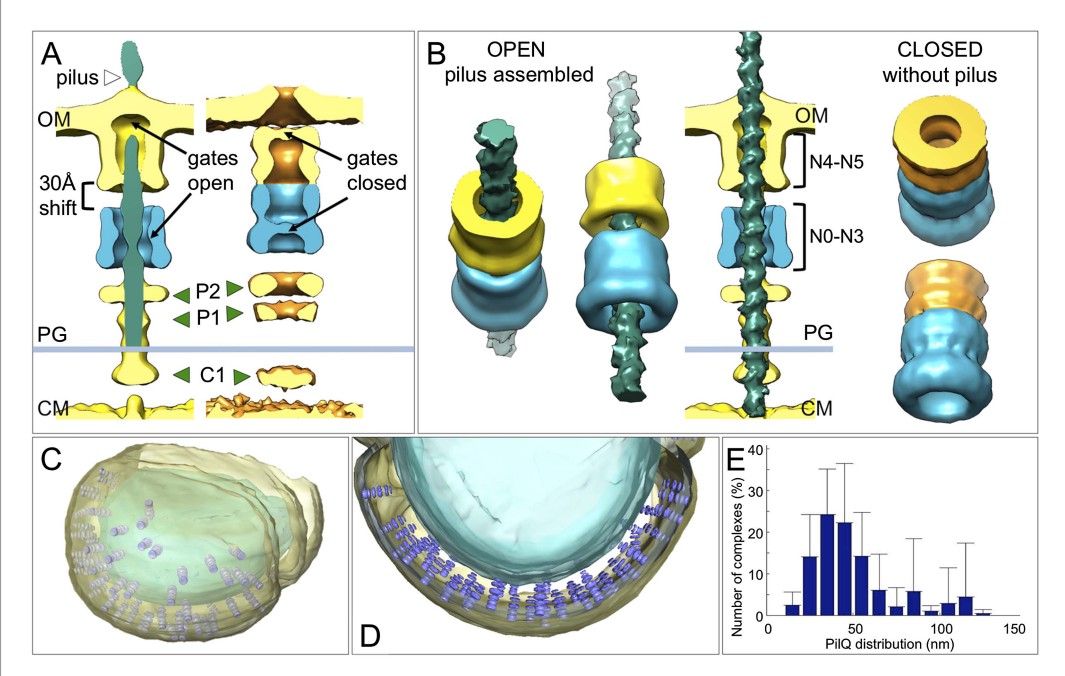

**Figure 5**. Changes between the open and closed state of the T4P machinery and its distribution *in situ*. (**A** and **B**) Comparisons between the PilQ components of the T4P machinery reveal large conformational changes whereby both gates open and domains N0–N3 (now shown in blue for both states) shift by ∼30 Å towards the cytoplasm on pilus extrusion. Green arrowheads indicate additional protein densities (C1 = proximal to the cytoplasmic membrane, P1 = central periplasmic ring 1, P2 = central periplasmic ring 2). In (**B**) the structure of the T4P has been docked into the open state. OM, outer membrane; PG, peptidoglycan; CM, cytoplasmic membrane. (**C** and **D**) Docking subtomogram averages (purple) into the tomographic volume of a cell reveals the distribution of the closed T4P machinery *in situ* with respect to the outer membrane (pale yellow) and cytoplasmic membrane (blue). See *Video 2* for details. (**E**) Averaged histogram of nearest-neighbour distance between protein complexes, calculated from 9 cells, with a total of 332 data points. Error bars indicate the standard deviation of the frequency distribution for each minimal distance.

The following figure supplement is available for figure 5:

**Figure supplement 1**. Subtomogram averages from cells with high periplasmic protein content.

machinery shown in all other figures were calculated from data collected at 33,000× and 7 µm underfocus on the Titan Krios microscope. Coordinates corresponding to the outer membrane and inner membrane domains of the complex were marked manually in IMOD (*Kremer et al., 1996*). Subvolumes from twice-binned tomograms were then extracted from NAD filtered (*Frangakis and Hegerl, 2001*) data and an initial alignment and averaging performed in SPIDER (*Frank et al., 1996*). This average was used as a reference for alignment and refinement using PEET (*Nicastro et al., 2006*). We have previously shown that *T. thermophilus* PilQ is a dodecamer by biochemical analysis (*Burkhardt et al., 2011*), which is supported by single-particle electron microscopy of *N. meningitidis* PilQ (*Collins et al., 2003*, *2004*). Therefore, 12-fold symmetry was applied to the core complex by 30° (360°/12 subunits) rotation of each subvolume prior to the alignment search. The final averages were obtained from 3984 particles for the closed complex and 312 particles for the open complex, using a mask drawn around PilQ. Any duplicates due to oversampling were removed in PEET (*Nicastro et al., 2006*). Due to the larger sample size, data for the closed complex were replaced by unfiltered tomograms. For 2D comparisons between states (*Figure 3C*, left panel and *Video 1*), the NAD filtered version with low-contrast noise removed is shown. For the pilus, 740 subvolumes of ∼2 nm length were selected in IMOD (*Kremer et al., 1996*), from unbinned CTF-corrected tomograms collected at 5 µm underfocus on the Titan Krios microscope. Subvolumes were aligned and averaged with a cylindrical mask and any duplicates due to oversampling were removed in PEET

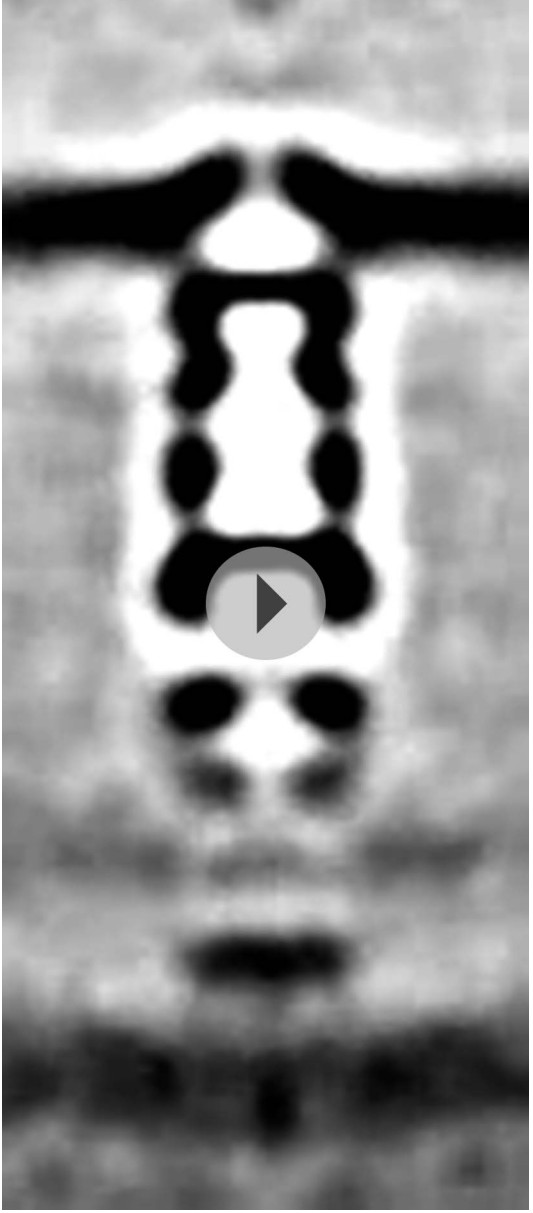

**Video 1.** Comparison of the T4P machinery in the open and closed state. The video was generated by morphing the two subtomogram averages in ImageJ.

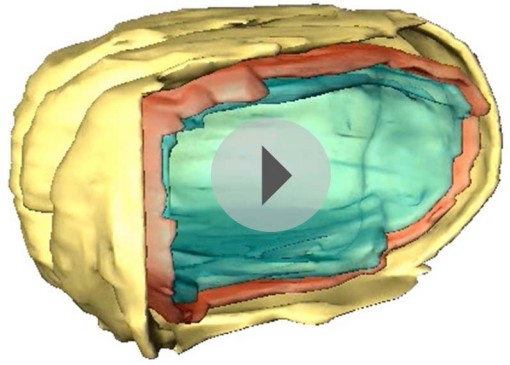

**Video 2.** Distribution of T4P complexes *in situ*. The video shows the rendered tomographic volume of a *T. thermophilus* cell. The outer membrane (pale yellow), peptidoglycan (orange), and cytoplasmic membrane (blue) are shown with the closed T4P complexes (purple).

(*Nicastro et al., 2006*). Resolution estimates were obtained using conventional 'even/odd' Fourier shell correlation (FSC), applying the 0.5 FSC criterion. A mask was drawn around the protein to exclude membrane and peptidoglycan from this estimate. The averages in *Figure 5—figure supplement 1* were smoothed by Gaussian filtering in UCSF Chimera, which was used to draw all surface views and remove low-contrast background noise using the 'hide dust' tool (*Pettersen et al., 2004*). The morph shown in *Video 1* was produced in ImageJ (*Schneider et al., 2012*). All subtomogram averages were uploaded to the EMDataBank (http://www.emdatabank.org) with ID codes 3021 (closed state, filtered), 3022 (closed state, unfiltered), 3023 (open state, filtered) and 3024 (pilus).

## Calculation of power spectra
Images of pili were cut out of a twice-binned tomographic slice and the unbinned subtomogram average (shown in *Figure 4C*) using helixboxer in EMAN2 (*Tang et al., 2007*). Power spectra were then calculated using the Iterative Helical Real Space Reconstruction (IHRSR++) software (*Egelman, 2007*).

### Calculation of the distribution of PilQ complexes in the closed state
The distance between PilQ complexes was determined with a MATLAB (Mathworks, California, USA) script (*Gold et al., 2014*). The centroid coordinates of complexes selected for subtomogram averaging were loaded into MATLAB and the distances calculated in an iterative for for-loop according to Pythagoras' theorem. This was performed for 332 closed PilQ complexes, combined from 9 different cells. Averaged histograms were calculated to depict the mean frequency of occurrence for each minimal distance. To account for the different numbers of PilQ complexes in each data set, the mean frequency was calculated as a percentage.

## Acknowledgements

We thank Bertram Daum for helpful comments on this manuscript, Deryck Mills for excellent maintenance of the EM facility, Andreas Walter for the MATLAB script, and Paolo Lastrico (Graphics Department, MPI of Biophysics, Frankfurt MPI) for assistance with *Figure 1*. This work was supported by the Max Planck Society and the Deutsche Forschungsgemeinschaft (AV 9/6-1).

## Additional information

### Competing interests

WK: Reviewing editor, *eLife*. The other authors declare that no competing interests exist.

### Funding

| Funder | Grant reference | Author |
| --- | --- | --- |
| Max-Planck-Gesellschaft (Max Planck Society) | | Vicki AM Gold, Werner Kühlbrandt |
| Deutsche Forschungsgemeinschaft (DFG) | AV 9/6-1 | Ralf Salzer, Beate Averhoff |

The funders had no role in study design, data collection and interpretation, or the decision to submit the work for publication.

### Author contributions

VAMG, Conception and design, Acquisition of data, Analysis and interpretation of data, Drafting or revising the article, Contributed unpublished essential data or reagents; RS, BA, Conception and design, Analysis and interpretation of data, Drafting or revising the article, Contributed unpublished essential data or reagents; WK, Drafting or revising the article, Contributed unpublished essential data or reagents

## Additional files

### Major datasets

The following datasets were generated:

| Author(s) | Year | Dataset title | Dataset ID and/or URL | Database, license, and accessibility information |
| --- | --- | --- | --- | --- |
| Gold VAM, Salzer R, Averhoff B, Kuehlbrandt W | 2015 | Structure of the type IV pilus machinery from Thermus thermophilus in the closed state | http://emsearch.rutgers.edu/atlas/3021_summary.html | Publicly available at EMDataBank (3021). |
| Gold VAM, Salzer R, Averhoff B, Kuehlbrandt W | 2015 | Structure of the type IV pilus machinery from Thermus thermophilus in the closed state | http://emsearch.rutgers.edu/atlas/3022_summary.html | Publicly available at EMDataBank (3022). |
| Gold VAM, Salzer R, Averhoff B, Kuehlbrandt W | 2015 | Structure of the type IV pilus machinery from Thermus thermophilus in the open state | http://emsearch.rutgers.edu/atlas/3023_summary.html | Publicly available at EMDataBank (3023). |
| Gold VAM, Salzer R, Averhoff B, Kuehlbrandt W | 2015 | Structure of the type IV pilus from Thermus thermophilus | http://emsearch.rutgers.edu/atlas/3024_summary.html | Publicly available at EMDataBank (3024). |

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
