## [Decision Letter]

Thank you for sending your work entitled “Structure of a type IV pilus machinery in the open and closed state” for consideration at *eLife*. Your article has been favorably evaluated by John Kuriyan (Senior editor) and three reviewers, one of whom is a member of our Board of Reviewing Editor and another Jan Löwe (peer reviewer).

The Reviewing editor and the other reviewers discussed their comments before we reached this decision, and the Reviewing editor has assembled the following comments to help you prepare a revised submission.

The short report titled “Structure of a type IV pilus machinery in the open and closed state” by Gold et al. describes the structure of a type IV secretion pore in two different states, bound and unbound to the pilus using cryo-electron tomography and sub-tomogram averaging. The message of the manuscript is straightforward. Although true molecular interpretation of the reported results will have to await studies at much higher resolutions, it is valuable to get an initial overview of what is happening in cells. The work presented is a very good example of what is currently possible.

A general concern of the reviewers was that only someone with considerable background in this field can follow the description and relate it to what is known from high resolution studies of components—e.g., the crystallographic studies the assembly platform (Karuppiah, et al.), secretion complex (Burckhardt, et al.), and fiber (Karuppiah, et al. and Craig, et al.). As the authors point out, the complex depicted in Figure 3 closely resembles that seen by Burckhardt, et al., but with some interesting structural differences. We therefore request a revised version that addresses the following points and that will allow a structural biologist, who does not know about bacterial pili, to understand the key points.

A) It would be helpful to use consistent terminology when introducing complexes and concepts. For example, the authors first introduce “secretins,” followed by reference to “the T4P system” and “the T4P machinery.” The molecular constituents of the T4P system/machinery/complex should be clearly delineated. Does T4P refer to the PilQ secretin complex alone (please see the Introduction), or is it meant to include both PilQ and PilMNO? Are PilMNO couplers, or parts of the complex? A cartoon of the model currently supported by the literature would be helpful.

B) Is the multi-ring structure, N0-N5, thought to be composed only of domains of the single protein (PilQ) or are there other components specific to particular rings. That is, do the authors think that the “gate closure” is a conformational change in PilQ itself or rather a redistribution of another protein?

C) Related to B). A major finding of the paper is a substantial conformational transition between the open and closed T4P machinery. (i) Since there are no atomic structures fitted into the cryoEM density presented, the importance of gate 2 in recognising the growing pilus is speculative. The entire paragraph should be reworded with more caution. (ii) The closed data set comes from EDTA-treated, pipetted, pilus-deficient cells, while the open set comes from non-pipetted, pilus-positive complexes. Comparison of the average number of pili to the average number of complexes suggests, as the authors write, that 80% of complexes are not associated with pili. Do these pilus-free complexes resemble the “closed” complexes seen in the pipetted, EDTA-treated cells?

D) To obtain the open state, the authors use strong EDTA treatment of cells to deplete the periplasm for the open state. This is an unnatural state for the pore proteins. Without this treatment, the signal to noise ratio is low in the tomograms, but the pore proteins can nonetheless be identified even without it (Figure 2). Did the authors attempt to carry out subtomogram averaging from these native data? What was the result? It would be interesting to know the result, even just from an image processing perspective.

E) Related to D). The authors should introduce a caveat in the Results section that some molecules associated with the secretion pore may have been removed by the EDTA treatment and not make such strong points about “in situ” structures: the periplasm has been shrunk and emptied, at least partially.

F) The authors comment that *T. thermophilus* was chosen due to its high transformation rate and convenient thermostability (Introduction). Could the authors comment on how their in situ findings might be generalized to other bacterial species that do not have unusually deep surface clefts or an “extraordinarily wide” periplasm (as stated in the Results and Discussion section)? Is it possible that these characteristics are responsible for exaggerated structural dynamics that more readily facilitate DNA uptake and thus the higher transformation rate?

G) Some slightly more technical points:

i) The IMOD reconstruction showing the pilus does not show strong features. It is hard to see repeats in Figure 4. Would it be possible to see a Fourier transform with a peak? The pili emanating from cells in Figure 2 are fairly long. Did the authors try to get 2D averages from the cryoEM images? If there is a repeat, then it should be easily visible from a few images. The issue is important because there are such strong differences from the pili of *N. gonorrhoeae*. In *Neisseria*, would one expect the T4P channel to be that much wider? Does it use other proteins or just more of them with more than 12-fold symmetry? Independent alignment and averaging of the present data should be tried, perhaps using the 0.143 FSC criterion.

ii) Is there any independent verification that T4P channels have 12-fold symmetry? The authors make heavy use of the symmetry, but it seems from the MS that they are relying on someone else's result.

iii) How confident can we be of the correspondence between the rings in the image from subtomogram averaging and the rings in the projection from negative staining of PilQ (Figure 3)? The spacings seem rather different. Could other components contribute to the rings in the intact sheath?

---

## [Author Response]

*A general concern of the reviewers was that only someone with considerable background in this field can follow the description and relate it to what is known from high resolution studies of components—e.g., the crystallographic studies the assembly platform (Karuppiah, et al.), secretion complex (Burckhardt, et al.), and fiber (Karuppiah, et al. and Craig, et al.). As the authors point out, the complex depicted in*
Figure 3
*closely resembles that seen by Burckhardt, et al., but with some interesting structural differences. We therefore request a revised version that addresses the following points and that will allow a structural biologist, who does not know about bacterial pili, to understand the key points*.

*A) It would be helpful to use consistent terminology when introducing complexes and concepts. For example, the authors first introduce “secretins,” followed by reference to “the T4P system” and “the T4P machinery.” The molecular constituents of the T4P system/machinery/complex should be clearly delineated. Does T4P refer to the PilQ secretin complex alone (please see the Introduction), or is it meant to include both PilQ and PilMNO? Are PilMNO couplers, or parts of the complex? A cartoon of the model currently supported by the literature would be helpful*.

We have unified the terminology throughout the paper to clarify that T4P machinery means the PilQ secretin plus PilMNO and other Pil proteins. We also include a new Figure 1, which is a schematic of the current model supported by the literature.

B) Is the multi-ring structure, N0-N5, thought to be composed only of domains of the single protein (PilQ) or are there other components specific to particular rings. That is, do the authors think that the “gate closure” is a conformational change in PilQ itself or rather a redistribution of another protein?

The multi-ring structure N0-N5 is composed only of dodecamers of PilQ. We know this from the comparison of our in situ structure to the 2D images of isolated PilQ, of which the dimensions (length and width of complex) are identical. In the previous 2D work, the protein in the sample was treated with hot phenol, resulting in dissociation into monomers of PilQ only with no other proteins present (4). Thus the gate closure is a conformational change in PilQ. We have made this point clearer in the text (Results and Discussion section).

*C) Related to B). A major finding of the paper is a substantial conformational transition between the open and closed T4P machinery. (i) Since there are no atomic structures fitted into the cryoEM density presented, the importance of gate 2 in recognising the growing pilus is speculative. The entire paragraph should be reworded with more caution*.

We appreciate that the function of the gates is speculative at present and have reworded the last four lines of the paragraph (third paragraph of the Results and Discussion section) accordingly to make this more general.

(ii) The closed data set comes from EDTA-treated, pipetted, pilus-deficient cells, while the open set comes from non-pipetted, pilus-positive complexes. Comparison of the average number of pili to the average number of complexes suggests, as the authors write, that 80% of complexes are not associated with pili. Do these pilus-free complexes resemble the “closed” complexes seen in the pipetted, EDTA-treated cells?

The closed data set is indeed obtained from EDTA-treated, pipetted and pilus-depleted cells. However, the open state is also EDTA-treated (as described in the Materials and methods, subsection headed “Sample preparation”) but not pipetted. The pipetting step produces shearing forces that destroy the pili. It was this alteration in sample preparation that preserved more pili on the cells and enabled structural determination. The average of the closed state was therefore calculated from pilus-depleted cells. This point is now made clear in Results. The value of 80% was calculated from the average number of pili (determined in detail by us elsewhere (46) and in agreement with tomographic data here), and the total number of closed complexes calculated from pilus-depleted cells. A comparison between open and closed complexes is explained in more detail in our response to D) below.

*D) To obtain the open state, the authors use strong EDTA treatment of cells to deplete the periplasm for the open state. This is an unnatural state for the pore proteins. Without this treatment, the signal to noise ratio is low in the tomograms, but the pore proteins can nonetheless be identified even without it (*Figure 2*). Did the authors attempt to carry out subtomogram averaging from these native data? What was the result? It would be interesting to know the result, even just from an image processing perspective*.

As explained above, both open and closed states were determined from EDTA-treated cells. EDTA treatment depletes the periplasmic protein to an extent that is difficult to control precisely. While most of our tomograms were collected from cells with low periplasmic protein level and correspondingly high contrast, we have also been able to average a few complexes from a tomogram of the two cells shown in Figure 2, where the periplasmic protein density was considerably higher. These cells contained complexes both in the open and closed state (with and without pili). The subtomogram averages are shown in the new Figure 5—figure supplement 1. Due to the poorer signal-to-noise ratio in the denser periplasm, we were able to identify reliably only a total of 16 open (the eye is guided by the pilus) and 11 closed complexes from these 2 cells combined. Even after Gaussian filtering to reduce noise the data quality is worse than for averages from cells with a more depleted periplasm. The new figure shows the PilQ component plus protein P2, which have distinguishable features. It is clear from these new data that the same large conformational changes occur in the T4P machinery, irrespective of the degree of periplasmic protein depletion, and hence of the effect of EDTA. Please note that without EDTA treatment, the periplasm was too dense to identify any complexes in the tomographic volumes.

*E) Related to D). The authors should introduce a caveat in the Results section that some molecules associated with the secretion pore may have been removed by the EDTA treatment and not make such strong points about “in situ” structures: the periplasm has been shrunk and emptied, at least partially*.

We have included the requested statement (Results and Discussion) and related it to the new subtomogram averages (Figure 5—figure supplement 1, and discussed in the fourth paragraph of the same section) as described above.

*F) The authors comment that* T. thermophilus *was chosen due to its high transformation rate and convenient thermostability (Introduction). Could the authors comment on how their in situ findings might be generalized to other bacterial species that do not have unusually deep surface clefts or an “extraordinarily wide” periplasm (as stated in the Results and Discussion section)? Is it possible that these characteristics are responsible for exaggerated structural dynamics that more readily facilitate DNA uptake and thus the higher transformation rate?*

*Thermus* thrives in hot environments. The wide periplasm (containing thick layers of secondary cell wall polymers) and the surface clefts may be an environmental adaptation. The T4P machinery does not co-localize with the clefts, so we do not think that the two are linked. It is however plausible that the length of the PilQ secretin has adapted to the wide periplasm. Importantly, proteins of the T4P machinery are functionally conserved (Nudleman et al. 2004) and thus we speculate that the overall architecture is similar across species. We have included a comment is this regard in the manuscript (at the end of the Results and Discussion section).

Whether PilQ in *Thermus* is responsible for particularly efficient DNA uptake and high transformation rates is questionable. Based on models of T4P-related DNA translocators, it is more likely that the AAA-ATPases or rudimentary DNA-translocator pili facilitate DNA uptake (Maier et al. 2002, [33]). As this point is rather speculative we prefer not to comment on it in the manuscript.

G) Some slightly more technical points:

*i) The IMOD reconstruction showing the pilus does not show strong features. It is hard to see repeats in*
Figure 4*. Would it be possible to see a Fourier transform with a peak? The pili emanating from cells in*
Figure 2
*are fairly long. Did the authors try to get 2D averages from the cryoEM images? If there is a repeat, then it should be easily visible from a few images. The issue is important because there are such strong differences from the pili of* N. gonorrhoeae*. In* Neisseria*, would one expect the T4P channel to be that much wider? Does it use other proteins or just more of them with more than 12-fold symmetry? Independent alignment and averaging of the present data should be tried, perhaps using the 0.143 FSC criterion*.

We have altered Figure 4 to a new tomographic slice that shows the repeat more clearly. We also now include a slice of the subtomogram average and the power spectra of both images. The power spectra show clear layer lines, which indicate that the structure is periodic. The distance of the layer lines from the equator is reciprocal to the pitch of the helix, which is measured as 1/(49 Å) or 4.9 nm. From the subtomogram average volume, we measured a repeat distance of 5.5 nm. Considering that the pixel size in the average is 0.42 nm, this change of 0.6 nm is within reasonable error of only ∼1.5 pixels. Calculation from the power spectra is less ambiguous, thus we especially thank the reviewers for this excellent suggestion. We have modified the repeat distance accordingly.

We did not try to get 2D averages from the cryoEM images because the signal from a single tilt series image (at low magnification and low dose compared to single particle cryo-EM images) is too weak to distinguish small features. We can barely see the pili in the images without performing a full tomographic reconstruction. Therefore we cannot align and average the present data by an independent method.

The most striking difference that can be observed at this resolution between the *N. gonorrhoeae* and *T. thermophilus* pili is their width. We have altered the Discussion and figure legend (Figure 4) in this respect. In the present work we have independently verified the width of the *T. thermophilus* pilus from two different samples. The first is from the subtomogram average of the open state, and the second is from the subtomogram average of the pilus alone. Both of these reveal an average width of ∼3 nm, only half the width of the ∼6 nm *N. gonorrhoeae* pilus. Pili from various species have been shown previously to vary in diameter by negative stain electron microscopy and modelling, and apart from the N-terminus, sequences are not well conserved (12). Thus it is not surprising that they exhibit structural differences at the cryoEM level. This point has been made clearer in the text (Results and Discussion). We have also improved Figure 5 by docking our structure of the T4P into the open state of the machinery.

A cryoEM structure of an isolated and therefore closed state of PilQ exists from *N. meningitidis* (closely related to *N. gonorrhoeae*), which also displays 12-fold symmetry (9, 10). The cryo structure shows that the channel is 9 nm wide, tapering to a point, and a homology model of the pilus (based on the *N. gonorrhoeae* pilus) is shown to be 6.5 nm wide (9). The *Thermus* closed channel tapers from 8 nm at the widest point to 4 nm at the narrowest point, changing to a more constant 7 nm in the open state (Figure 3 and Figure 4). Therefore it is apparent that very large conformational changes allow the width of the channel to alter considerably on pilus extrusion. Thus we see no reason why the *Neisseria* channel formed by PilQ could not accommodate a pilus of 6 – 6.5 nm. We have made this point clearer in the text (Results and Discussion).

*ii) Is there any independent verification that T4P channels have 12-fold symmetry? The authors make heavy use of the symmetry, but it seems from the MS that they are relying on someone else's result*.

We do make use of 12-fold symmetry, but this is in fact based on our own published work (4), which is also supported by others (9, 10). We did not think it was necessary to verify the symmetry again in this work and have made the point clearer in Materials and methods (in the subsection headed “Subtomogram averaging”).

*iii) How confident can we be of the correspondence between the rings in the image from subtomogram averaging and the rings in the projection from negative staining of PilQ (*Figure 3*)? The spacings seem rather different. Could other components contribute to the rings in the intact sheath?*

The spacing between the rings is slightly different in the detergent-solubilized stained structure, and in situ in cells. However the complexes are in very different environments. The in situ environment has been cryo-preserved and the complexes, presumably with all their accessory proteins, are still embedded in the membrane, whereas the isolated PilQ is detergent-solubilized and stained. It is not surprising that a large complex that can undergo such major conformational changes may look slightly different after detergent solubilization and air-drying in negative stain. Nevertheless, whilst the ring spacing appears marginally altered, the dimensions of the subtomogram average of PilQ (diameter of complex, diameter of pore and length of complex) perfectly matches that of isolated PilQ. We have now included more of these comparative dimensions in Figure 3 to make this point clear. Therefore, as also explained in point B), we do not think that other proteins contribute to the rings that have been assigned to PilQ.